# The study on the adsorption characteristics of anthracite under different temperature and pressure conditions

Danan Zhao [ID]*, Xiaofei Ke, Mincong Huang, Wenchang He, Mingyu Tong, Baihong Chen, Qu Du

Faculty of Civil Engineering and Architecture, Zhanjiang University of Science and Technology, Zhanjiang, Guangdong, China

* 2373562966@qq.com

## Abstract

The study of the adsorption characteristics of coal is of great significance to gas prevention and $CO_2$ geological storage. To explore the adsorption mechanism of coal, this study focuses on columnar anthracite. Adsorption tests on coal rock under a range of physical field conditions were conducted using the volumetric method. The adsorption characteristics of anthracite for $CO_2$, $CH_4$, and $N_2$ gases under different conditions were investigated using Grand Canonical Monte Carlo (GCMC) and Molecular Dynamics (MD) methods. The results showed that the adsorption capacities of anthracite for these three gases are in the order of $CO_2 > CH_4 > N_2$, and that the adsorption capacity increases with increasing gas injection pressure. The $CO_2/CH_4/N_2$ gas molecule adsorption capacity of the anthracite macromolecular structure model decreases with increasing temperature. The increase in temperature has the greatest influence on the $CO_2$ absorption capacity, followed by the $CH_4$ and $N_2$ adsorption capacities. The research offers a theoretical basis for the control of coal mine gas and the geological storage of $CO_2$.

## 1. Introduction

As the main energy source in China, coal has provided strong support for social and economic development and will continue in its role as a ballast stone in maintaining stable economic operations for a long time in the future [1,2]. China is endowed with a substantial reserve of anthracite, representing 14% of the country's total coal resources. As a crucial raw material for deep coal processing, anthracite has been instrumental in the early stages of China's coal chemical industry. Concurrently, a considerable volume of coalbed methane resources is present within anthracite, yet incidents of coal mine gas disasters persist [3,4]. The principal reason for this is that research into the development theory and technology of coalbed methane ($CH_4$, $CO_2$, $N_2$ and other gases) is relatively limited. In particular, there is still a lack of understanding of the mechanism of adsorption characteristics of coalbed methane, which is resulting in an unsatisfactory development effect of coalbed methane. Accordingly, this study employs anthracite as the experimental coal sample to investigate the adsorption

**Data availability statement:** All relevant data are within the manuscript and its Supporting Information files.

**Funding:** This work was partly supported by the Scientific Research Project of Guangdong Provincial Department of Education—Young Innovative Talents Project (grant number 2022KQNCX141).

**Competing interests:** The authors have declared that no competing interests exist.

characteristics of coalbed methane in the reservoir, which represents a critical and pressing issue in the development and research of coalbed methane [5,6].

The adsorption capacity of coal to gas is affected by many influencing factors, which can be roughly divided into two aspects—(i) the nature of coal itself and (ii) external factors. The nature of coal mainly refers to the macroscopic composition of coal and the degree of coal metamorphism. In comparison, external factors include gas composition, coal moisture, temperature, and particle size [7–9]. Ranathunga [10] studied the applicability of $CO_2$ to enhance coalbed methane recovery through $CO_2$-driven coalbed methane experiment, and the results showed that $CO_2$ injection had a higher coalbed methane recovery rate than natural extraction, and effectively improved the anti-reflection performance of coal seams. Joubert et al. [11] studied the effect of moisture content on the adsorption behavior of $CH_4$ gas in four different bituminous coals at 30°C and 6 MPa. It was found that moisture inhibited the adsorption capacity of coal, and that higher oxygen-containing functional group contents in coal strengthened the inhibition of moisture on its adsorption capacity. However, when the moisture content exceeded a certain value, no further effect was observed on the adsorption capacity of $CH_4$. Pini et al. [12] performed coal adsorption and expansion experiments with He, $CO_2$, $CH_4$, and $N_2$. The results showed that the expansion degree of coal followed the trend: $CO_2 < CH_4 < N_2 < He$. The He gas had a weak adsorption capacity with an almost negligible volume change. A substantial body of research has been conducted by scholars in the field of coal adsorption capacity. However, the constraints imposed by experimental conditions have resulted in the inability to derive more than macroscopic laws from the observed phenomena.

Many factors affect the adsorption characteristics of coal, and the microscopic adsorption mechanism cannot be fully revealed through experimental methods. Therefore, many scholars have used molecular simulation methods and experiments to research these topics, which can make up for the defects of experiments and can effectively reveal the influence mechanism of the gas adsorption performance [13–15]. Wang et al. [16] analyzed the effect of high-temperature and high-pressure conditions on the adsorption characteristics of coking coal using isothermal adsorption experiments and the molecular dynamics method, and they established an isothermal adsorption model suitable for high-temperature and high-pressure conditions. It was proven that the molecular dynamics method can better characterize the adsorption characteristics of coking coal. Based on the first-principles calculation of density functional theory. Zeng [17] simulated the coal-bed methane mining process using the adsorption-strain coupling model, and the results showed that the microsimulation could explore the changes in coal deformation and permeability that were difficult to find in experiments. Hou et al. [18] used the Materials Studio (MS) molecular simulation software to study the competitive adsorption of $CH_4$ and $CO_2$ and found that the adsorption capacity of montmorillonite decreased with increasing temperature, initially increased and then decreased with increasing pressure, and decreased with increasing pore size. To study the occurrence of methane in micropores, Han et al. [19] constructed coal structure models with different pore sizes and found that the adsorption capacity was positively correlated with the pore size of pure gas adsorption. Mosher et al. [20] used the MS molecular simulation software to study the effect of pore size on adsorption behavior. They compared their numerical simulation results with experimental data, and the molecular simulation played an important role in determining the accurate capacity below the nanometer scale. Fitzgerald et al. [21] established a simplified local density model to predict the adsorption behavior of different gases. Their results revealed that the model can predict the adsorption behavior of $CH_4$, $N_2$, and $CO_2$ on coal and considers the influence of adsorption surface structure characteristics on the adsorption behavior, which can improve the ability to predict high-pressure adsorption phenomenon. As evidenced by

the preceding analysis, research has been conducted on the adsorption characteristics of coal seams [22–24]. Nevertheless, the majority of scholars have focused their research on pulverised coal, with relatively few studies examining columnar coal, which is in closer proximity to the coal storage environment. Additionally, there is a paucity of research examining the microscopic mechanisms of adsorption through the integration of molecular simulation and experimentation.

Given this, in this paper, a coal rock adsorption test platform under multi-physical field conditions is applied to explore the variations in and controlling mechanism of the coal adsorption capacity with adsorption pressure during the adsorption of $CO_2$, $CH_4$, and $N_2$ by columnar anthracite from the Yangquan Coal Mine, Shanxi Province, China, under stress conditions. Based on experimental research and through molecular dynamics simulation, the pore structure characteristics of an anthracite macromolecular structure model and the influence of temperature on the adsorption performance of the anthracite molecular structure model for single component $CO_2$, $CH_4$, and $N_2$ gases were studied from the microscopic point of view. The research findings presented in this paper are valuable for enhancing our understanding of the adsorption characteristics and mechanisms of $CH_4$, $CO_2$, and $N_2$ in coal. The aim is to elucidate the influence of factors such as adsorption pressure, temperature, gas properties, and other variables on the molecular-level adsorption and diffusion behaviors of gases. This study provides theoretical support and technical guidance for optimizing $CH_4$ treatment and $CO_2$ geological storage in coal mines.

## 2. Experiment and simulation

### 2.1. Experimental design and procedures

This experiment aimed to study the change rule and controlling mechanism of the $CH_4$, $CO_2$, and $N_2$ adsorption capacities and equilibrium pressures of anthracite. The pillar coal used in the experiment was anthracite collected from a mine in eastern Shanxi. Using a coal rock adsorption test platform under multi-physical field conditions, the pure $CH_4$, $CO_2$, and $N_2$ adsorption tests on the anthracite were performed using the equal volume method (Fig 1) [25]. According to the geological conditions at the coal sample collection site, the experimental temperature was finally set to 30°C, and an axial pressure of 10 MPa was applied to the

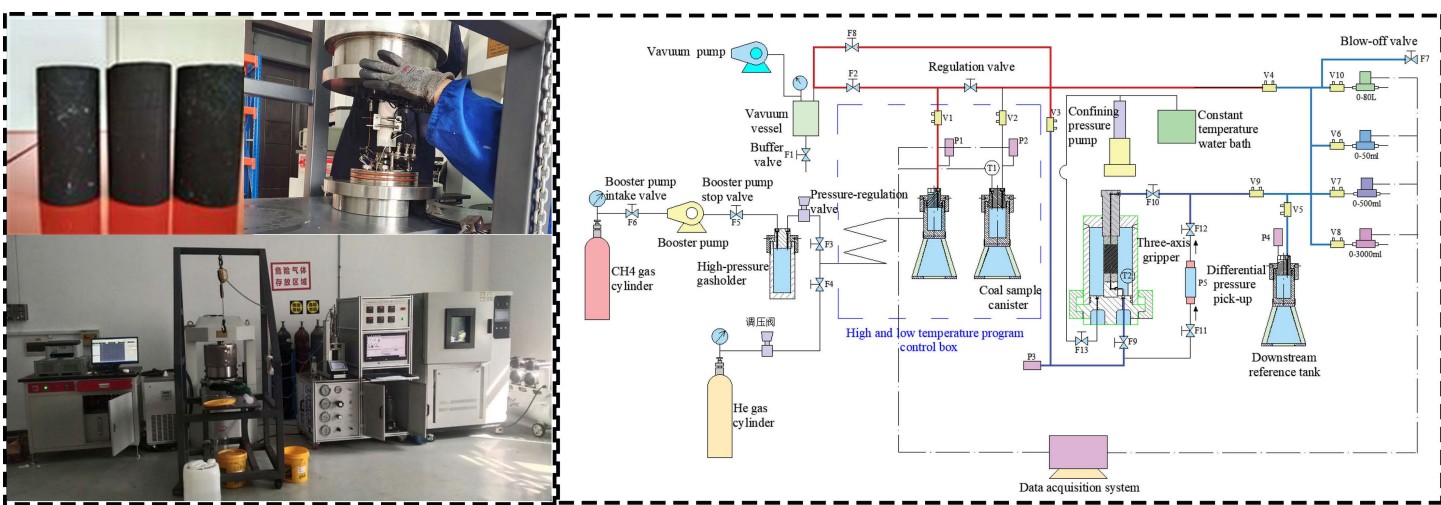

**Fig 1. Schematic of the experiment apparatus.**

tested coal sample. Furthermore, a confining pressure of 8 MPa was applied, and the adsorption equilibrium pressure was set at seven points, ranging from 0.5 to 6 MPa. The experimental test steps are as follows:

**Installation of specimen.** An anthracite standard specimen with a size of φ50 mm × 100 mm (±0.1 mm) was installed on the base of the pressure chamber, and then, a sensor was installed on the top of the coal sample. Then, the shell was closed, and the valve at the oil injection point of the pressure chamber was tightened. After the oil injection, the confining pressure required for the test was adjusted using the confining pressure control system.

**Vacuum pumping.** After the coal sample was installed and after the system temperature reached the preset value, vacuum valves F2 and F8 were opened to vacuum the system to discharge any impurities such as air and water from the entire device.

**Free volume determination.** The free space volume ($V_f$) measurement of the system is one of the key factors to ensure the accuracy of the experimental results. The water bath temperature was set to 30°C, after which the He cylinder was opened at valves $V_1$ and $V_3$. An upstream reference tank allowed the entry of 0.3 MPa the He gas, after which $V_1$ was closed until the pressure sensor shows the pressure value was stable; the pressure at $P_{10}$ at then recorded. Valve $V_3$ was then opened and He in the reference tank began to flow into the pressure chamber. When the pressure of the reference tank and the pressure cabin was balanced, the pressure value $P_{11}$ at this time was recorded, and the volume of the first free space was obtained.

After the first test, the injection gas pressure was continuously increased until the three tests were completed; the average values of $V_{f1}$, $V_{f2}$ and $V_{f3}$ were taken. Finally, the system automatically calculated the free volume. After the free volume test was completed, the system was vacuumed again to ensure that the He gas inside the system and the coal sample was completely pumped out to avoid experimental errors.

**Adsorption experiment.** In this experiment, the experimental test of adsorption of pure $CH_4$, $CO_2$ and $N_2$ by anthracite coal was carried out by the isovolumetric method. According to the burial depth of the experimental coal samples, the reservoir pressure, the temperature and the distribution of the geostresses, the final setup of the experimental temperature was 30°C. The axial pressure of the test samples was applied at 10 MPa, the enclosing pressure was applied at 8 MPa, and the equilibrium pressure of the adsorption was taken as 7 points, ranging from 0.5 to 6 MPa. For the convenience of the calculations, the effective stress is defined as: 1/ 3 × [Axial pressure + 2 × Confining pressure] -Equilibrium pressure. The isothermal adsorption experiment interface was opened, the experimental gas was selected, the Peng-Robinson (PR) equation of state was selected, the equilibrium judgment time was set, and the pressure fluctuation range was set to 0.05 MPa (up to 14 pressure numbers could be set). The run was started. First, vacuum was achieved according to the system prompt, and then, gas was added to the reference tank to reach the first test pressure. After the gas was stable, the system automatically opened pneumatic valve $V_2$ to achieve adsorption equilibrium, and it automatically measured the adsorption volume and adsorption amount. After completing the first test pressure, the gas was added to the reference tank to reach the second test pressure value. Once the gas was stable, the system automatically measured the adsorption volume and adsorption capacity under the second pressure condition. By analogy, when the adsorption experiment began, the current storage data table was opened simultaneously, and the storage cycle was set. The system automatically saved the temperature, pressure, and other data for the coal sample tank and the reference tank during the experiment. When the adsorption of each target pressure coal sample reached equilibrium, the system automatically calculated the amount of adsorption as follows:

$$Q_i = \frac{V_m \left( \dfrac{P_{i0} V_{us}}{Z_{i0}} - \dfrac{P_{i1} \left( V_{us} + V_f \right)}{Z_{i1}} \right)}{mRT},$$

(1)

where $V_m$ is the molar volume of the gas (22.4 L/mol); $P_{i0}$ is the initial pressure of the ith upstream reference tank (MPa); $V_{us}$ is the system automatic test upstream reference tank and pipeline volume; and $V_f$ is the volume of the free space. Additionally, $Z_{i0}$ is the initial compression factor of the ith gas, $P_{i1}$ is the equilibrium pressure of the gas in the ith reference tank (MPa); $Z_{i1}$ is the compression factor of the ith equilibrium gas; R is the gas molar constant; and T is the experimental temperature (K).

## 2.2. Model construction

To make up for the shortcomings of the experiment, in this study the MS molecular simulation software was used to study the microscopic mechanism of the adsorption behavior of the anthracite macromolecular model on coalbed methane ($CH_4$, $CO_2$, and $N_2$) molecules. In this paper, according to the basic parameters of the anthracite coal samples used in the experiment, a self-built three-dimensional model of anthracite molecules is cited (Fig 2a). The rationality of the model is verified by comparison of experiments and gas adsorption analysis [26]. Using the amorphous cell module, the calculation accuracy was fine, the force field was the condensed-phase optimized molecular potentials for atomistic simulation studies (COMPASS). Then, the geometry optimization module was used to optimize the structure of the crystal cell model of anthracite. The COMPASS force field was selected, and the kinetic optimization of the anneal module and the dynamics module was further processed [27,28]. The purpose was to minimize the energy of the constructed coal molecular structure model and stabilize it and to obtain the low-energy conformation of the anthracite macromolecular model.

The macromolecular structure model and micropore distribution of the anthracite model are shown in Fig 2b and 2c. The adsorption mechanism of the single component $CO_2$, $CH_4$, and $N_2$ gas molecules on the anthracite macromolecular structure model was analyzed using the grand-canonical Monte Carlo and molecular dynamics methods. The gas adsorption simulation was performed using the fixed pressure task in the sorption module. Set the radius of Connolly probe to 0.13nm (molecular dynamics radius of He), the mesh resolution to 0.75 Å, Equilibration steps and Production steps to 1000000. The COMPASS force field was selected (this force field is suitable for simulating the adsorption of small gas molecules such as $CO_2$ and $CH_4$ in coal macromolecules) [29,30].

## 3. Results and discussion

### 3.1. Analysis of isothermal adsorption experiment results

The isothermal adsorption curve is particularly important for the evaluation of coalbed methane reserves and development potential. The recoverable amount of coalbed methane can be predicted based on the isothermal adsorption curve of coal. The critical desorption pressure of coalbed methane can be determined to assess coalbed gas storage and saturation [31]. The adsorption isotherms of $CO_2$/$CH_4$/$N_2$ on anthracite are shown in Fig 3.

The experimental results show that the $CO_2$/$CH_4$/$N_2$ adsorption capacity of anthracite increased with increasing gas injection pressure, mainly due to the expansion effect of coal after gas adsorption, which may have increased the pore size of the micropores in

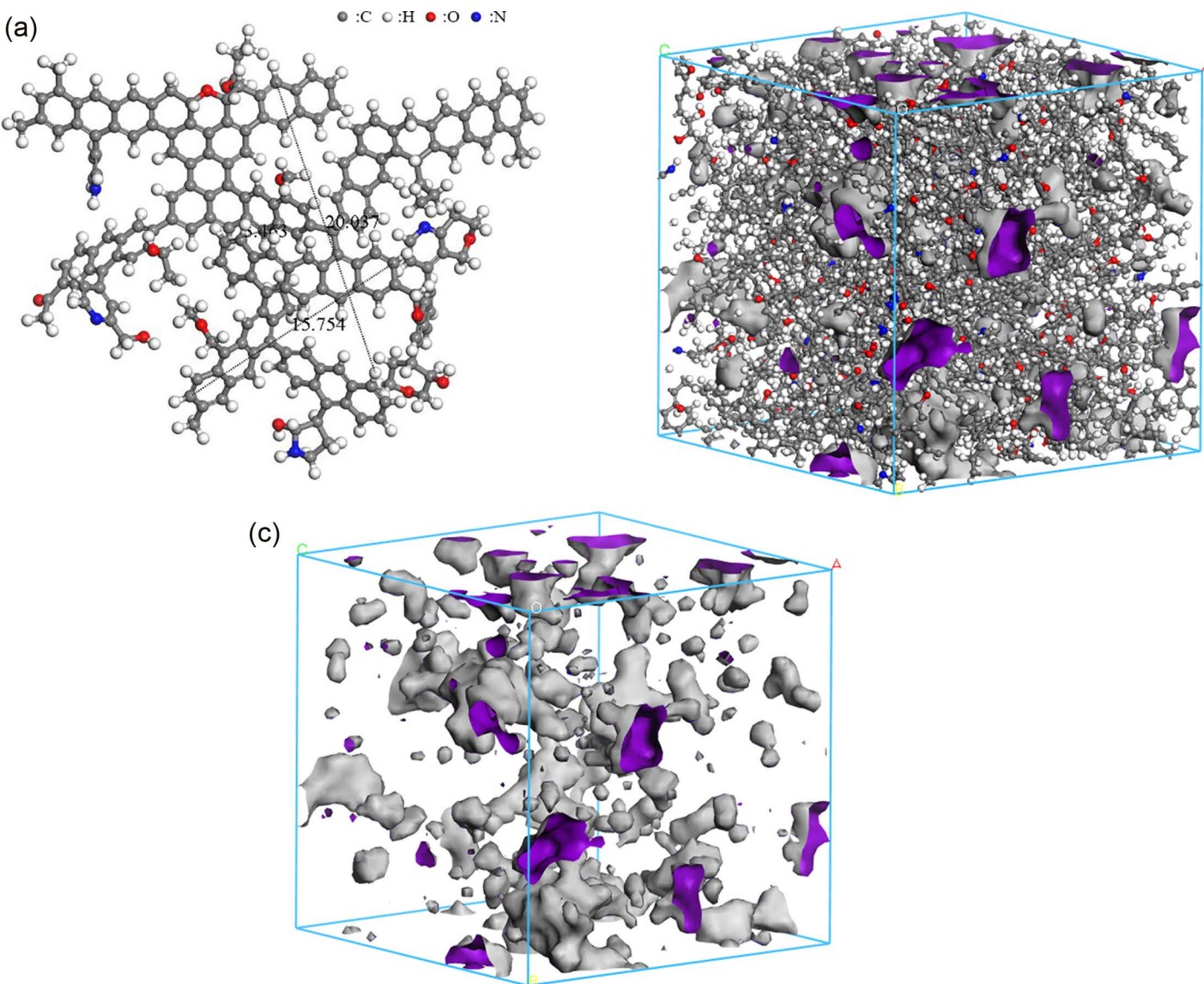

**Fig 2. Numerical model of anthracite.** (a) Three-dimensional model of anthracite (b) Crystal cell model (c) Micropore distribution.

the coal, increasing the gas adsorption capacity of anthracite. Under low pressure (P < 3 MPa), the $CO_2/CH_4/N_2$ adsorption capacity increased relatively quickly, and the isothermal adsorption curve was steeper. When the pressure was higher (P > 3 MPa), the amount of $CH_4/N_2$ gas adsorbed on the anthracite increased relatively slowly, and the amount of adsorption tended to be saturated. This was because in this stage, most of the adsorption sites on the coal surfaces were occupied by gas, and the remaining adsorption sites were less abundant, which led to a decrease in the rate of increase in the amount of adsorption. However, the amount of $CO_2$ adsorption exhibited a sharp increase at approximately 4 MPa. This was mainly due to the differential expansion effect of the coal on the adsorption of $CO_2/CH_4/N_2$. When it was at the critical pressure point of gas injection, the rate of increase in the density of $CO_2$ was faster than those of $CH_4$ and $N_2$, resulting in expansion of the coal seam after $CO_2$ injection.

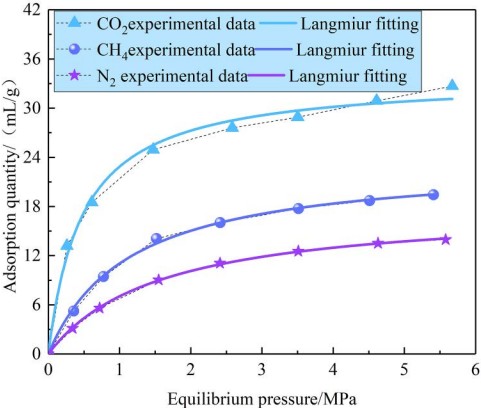

**Fig 3. Isothermal adsorption curves of $CO_2$/$CH_4$/$N_2$ adsorbed by anthracite coal.**

The gas adsorption capacities of anthracite were $CO_2$ > $CH_4$ > $N_2$. The adsorption capacities of the coal for the different gases were different because of the different physical and chemical properties of the three gas molecules and the different interactions between the gas molecules and coal surfaces. The physicochemical properties of $CO_2$, $CH_4$, and $N_2$ are listed in Table 1. In the process of gas adsorption by coal, the gas itself needs to lose part of its energy to adsorb onto the surfaces of coal molecules, and the part of the energy lost is closely related to the polarizability of the gas molecule. The larger the polarizability of the gas molecule is, the greater the dispersion force and induction force are. This makes it easier for the gas to occupy the adsorption site on the surface of the coal, which is conducive to the adsorption of gas by coal [32]. Under the joint action of comprehensive factors, anthracite has the strongest $CO_2$ absorption, followed by $CH_4$ and $N_2$.

Fig 3 indicates that the isothermal adsorption curve of coal can described by the Langmuir equation:

$$Q = \frac{abP}{1+bP} \tag{2}$$

where $a$ is the saturated adsorption capacity of $CO_2$/$CH_4$/$N_2$ gas, $mL \cdot g^{-1}$; and $b$ is the reciprocal of Langmuir pressure, $MPa^{-1}$.

The $CO_2$, $CH_4$, and $N_2$ adsorption isotherms of the coal samples conformed to the Langmuir equation, and the fitting equation coefficients were 0.9796, 0.9967, and 0.9982, respectively. Based on the Langmuir equation, the effective stress in the experimental data was fitted with the amount of adsorption, and the variation in the gas adsorbed by the coal sample under the effective stress was obtained (Fig 4).

**Table 1. Physicochemical properties of $CO_2$, $CH_4$ and $N_2$.**

| Gas mode | $CO_2$ | $CH_4$ | $N_2$ |
|---|---|---|---|
| Dynamic diameter(nm) | 3.30 | 3.80 | 3.64 |
| Critical temperature(K) | 304.29 | 190.45 | 126.15 |
| Critical pressure(MPa) | 7.38 | 4.57 | 3.40 |
| Boiling point(K) | 194.65 | 111.55 | 77.35 |
| Polarizability ($10^{-25}$ $cm^3$) | 26.50 | 26.00 | 17.60 |

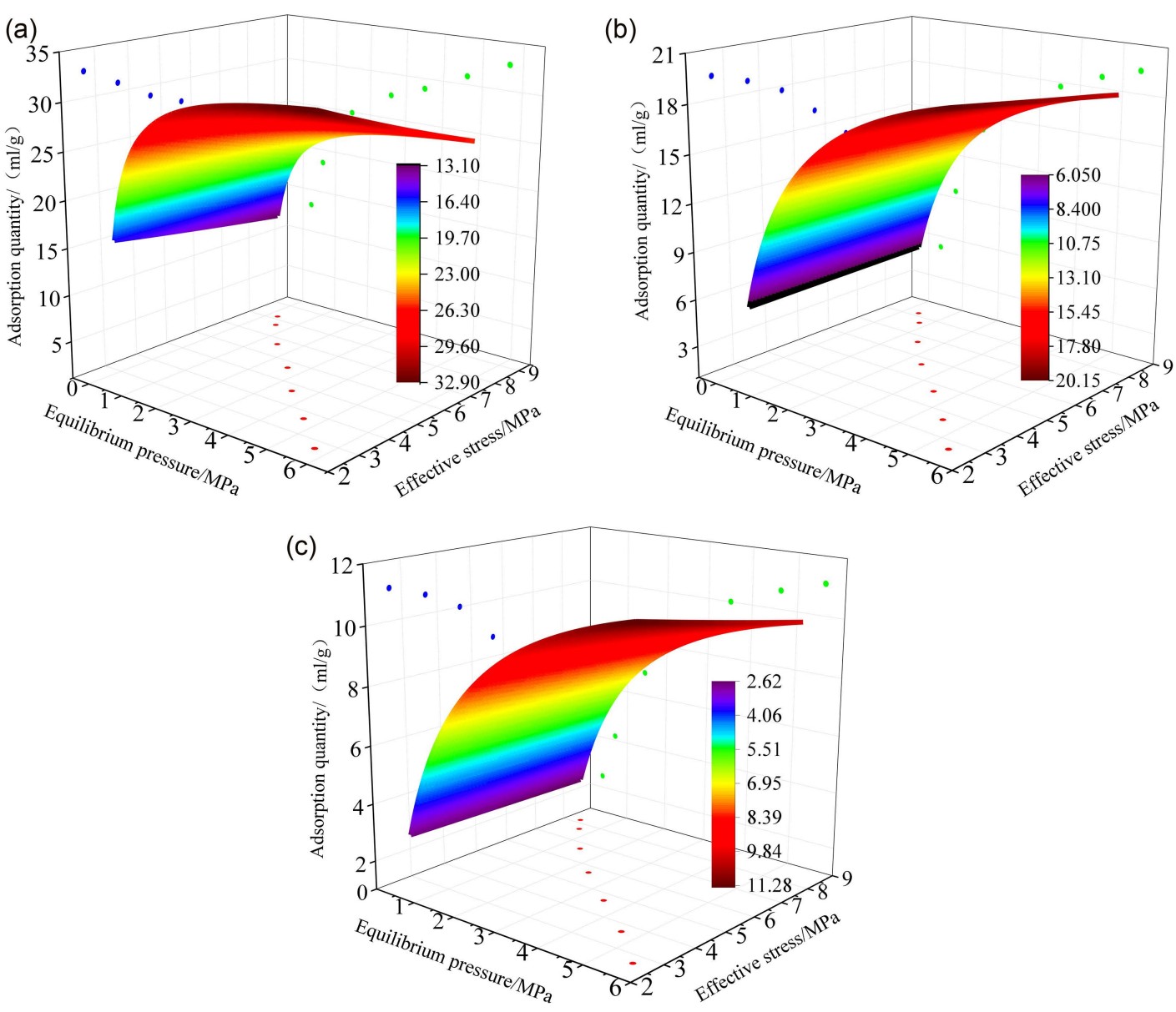

**Fig 4. Variation in the gas adsorbed by the coal samples under effective stress.** (a) $CO_2$ gas (b) $CH_4$ gas (c) $N_2$ gas.

It can be seen from Fig 4 that the $CO_2$, $CH_4$, and $N_2$ adsorption capacities of anthracite decreased with increasing effective stress. This was because the pore size in the coal rock changed with the gas pressure and the stress acting on the coal skeleton, that is, the effective stress control. The increase in the effective stress led to the closure of some of the micropores, so the pore size was smaller than the critical pore scale. Furthermore, it was difficult for the gas to enter this part of the pores, which led to a decrease in the saturated adsorption capacity of the coal rock. Moreover, the increase in the effective stress led to a decrease in the porosity, resulting in a decrease in the surface area of coal and loss of surface adsorption sites, which also led to a decrease in the saturated adsorption. In addition, the fitting results indicated that the increase in the effective stress had the strongest inhibition effect on the $CO_2$ adsorption capacity, followed by those of $CH_4$ and $N_2$.

### 3.2. Analysis of simulation results of pore structure characteristics

The pore volume and specific surface area of the coal macromolecular structure are not fixed, and the pore characteristics are closely related to the molecular radius of the Connolly probe [33,34]. The smaller the radius of the probe molecule is, the larger the pore volume that can be detected is. The gases used in this study were $CO_2$, $CH_4$, and $N_2$, and the probe radius was equivalent to the molecular dynamics radii of the three gases, namely, 0.165 nm, 0.19 nm, and 0.182 nm. Due to the different kinetic radii of $CO_2$, $CH_4$, and $N_2$ molecules, the pore volume and specific surface area that could be detected in the same coal macromolecular structure were also different, resulting in different pore information being obtained, which made the calculated pore distribution characteristics very different (Fig 5).

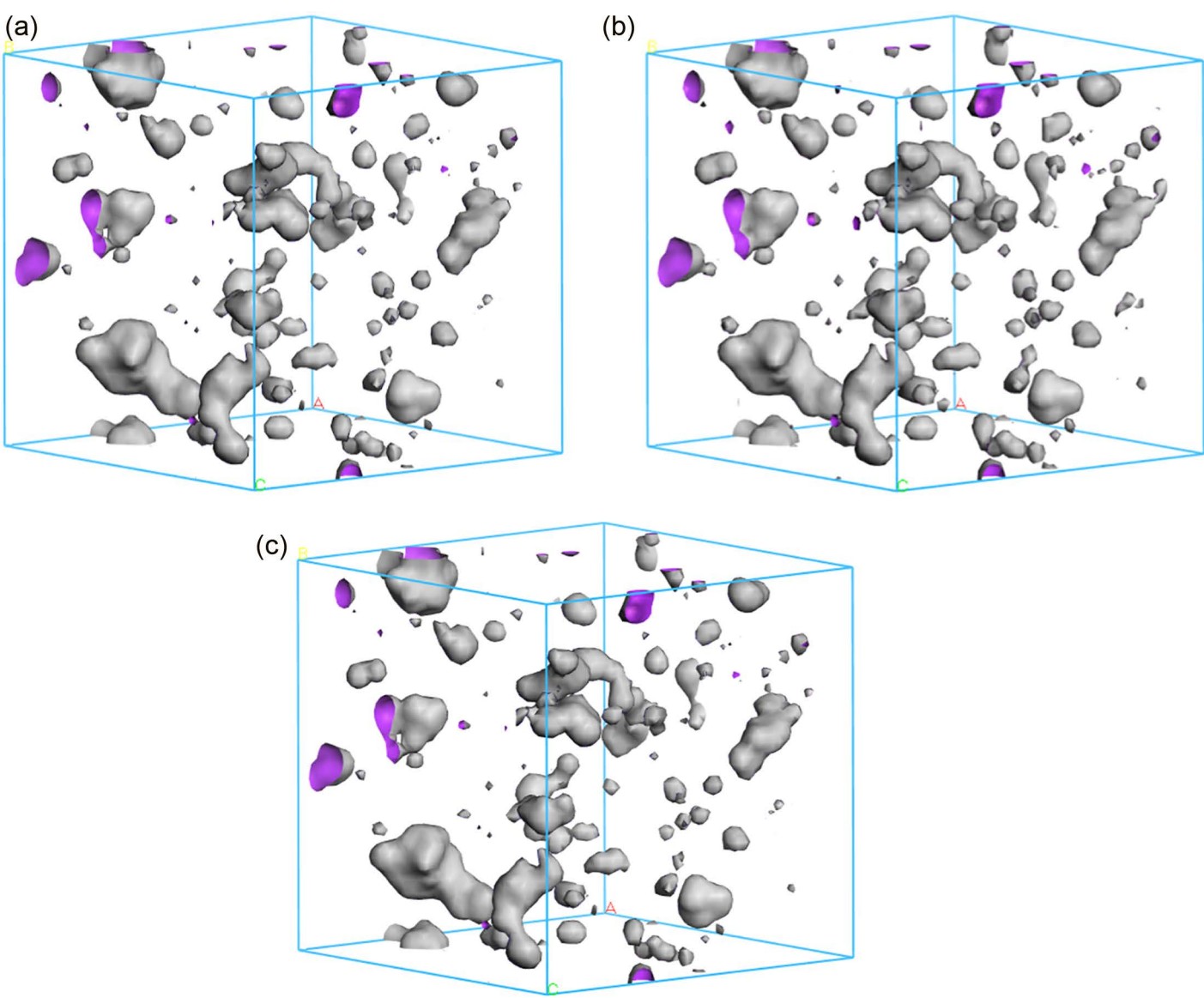

**Fig 5. Micropore distribution of $CO_2$, $CH_4$, and $N_2$ on the anthracite molecular structure.** (a) $CO_2$ (b) $CH_4$ (c) $N_2$.

In the macromolecular structure of anthracite, the micropore volume detected for $CO_2$ was 2975.78 $Å^3$/g, and the specific surface area was 3787.95 $Å^2$/g. The micropore volume detected for $CH_4$ was 2118.40 $Å^3$/g, and the specific surface area was 2705.77 $Å^2$/g. The micropore volume detected for $N_2$ was 2327.42 $Å^3$/g, and the specific surface area was 3021.81 $Å^2$/g. In the macromolecular structure of anthracite, the micropore volume and specific surface area detected for the $CO_2$ molecules were significantly higher than those for $CH_4$, followed by those for $N_2$. This explains why some micropores in the coal body can be detected using $CO_2$ molecules but not using $CH_4$ and $N_2$ molecules. Similarly, for the same micropore, the space detected using $CO_2$ molecules is larger than that detected using $CH_4$ and $N_2$ molecules. In terms of the adsorption space provided by the micropores in the coal body, the $CO_2$ gas was more easily adsorbed by the coal body than the $CH_4$ and $N_2$ gas.

To further explore the pore distribution characteristics of the anthracite macromolecular structure, based on obtaining the pore distribution characteristics, it was sliced, and the atom volumes and surfaces module was used to continue to analyze the distribution of the entering and non-entering pores. In this study, the dynamic radius of $CO_2$ was 1.65 nm as was the molecular radius of the Connolly probe, and the macromolecular model of anthracite was used as the cutting object. The cutting position is shown in Fig 6. From bottom to top, slices 1–3 are shown. The distributions of the entering and non-entering pores in the macromolecular structure of anthracite are shown in Fig 7, in which the upper picture shows the accessible pores, and the lower picture shows the accessible and inaccessible pores.

Based on the slice results of the micropore pore distribution at different positions in the anthracite macromolecular model, it was found that the pores in anthracite were anisotropic, which reflects the complexity and disorder of the coal as a material with an amorphous structure. From the slices at different positions in anthracite, it was found that the accessible pores were mainly distributed at the edge of the slice, that is, around the macromolecular structure of the coal, and were relatively independent, simple in shape, small in number, and approximately ink bottle-shaped. The inaccessible pores were mainly distributed in the center of the slice, that is, in the interior of the coal macromolecular structure. Their shape was more complex, their number was greater, and the direct connectivity between the pores was better. In addition, the number of pore throats (purple area) that allowed the gas to enter the pores was small, and their diameter was small. This led to some of the gases with larger kinetic molecular radii being unable to enter the pores, which was not conducive to gas enrichment.

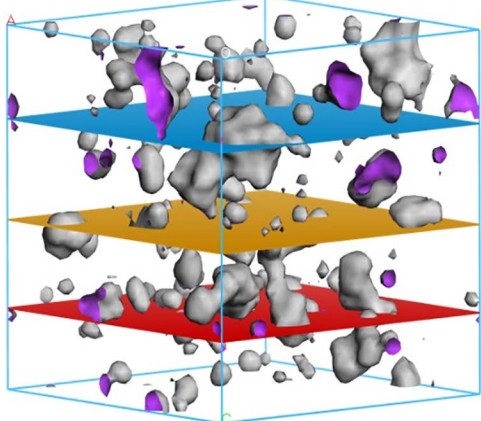

**Fig 6. Position of each slice of anthracite.**

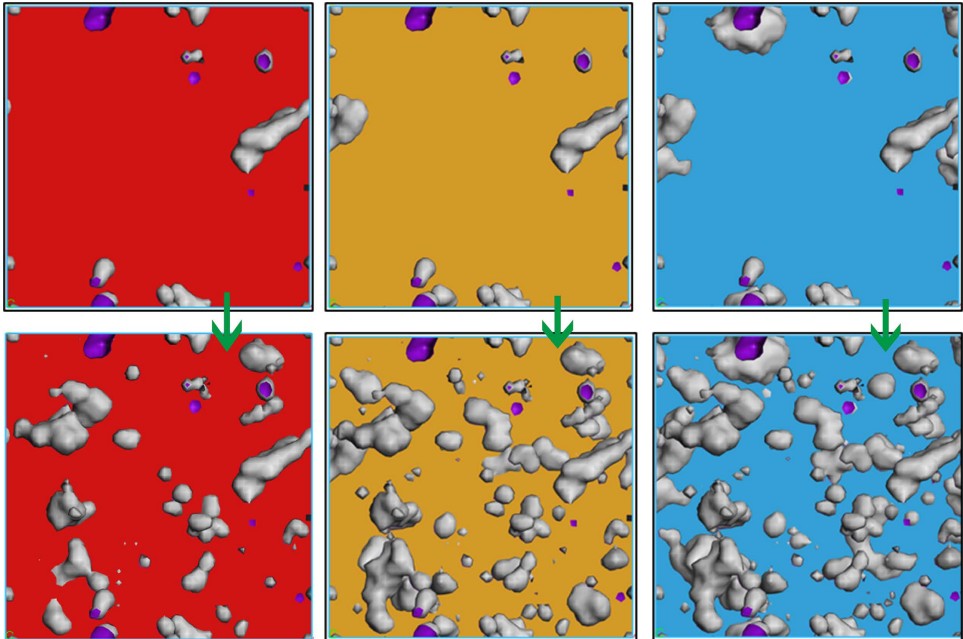

**Fig 7. Comparison of accessible and inaccessible pores in anthracite.**

Through analysis of the pore distribution characteristics of the anthracite macromolecular model, the characteristics of the micropore morphology, distribution, connectivity, and heterogeneity of the anthracite macromolecular structure were revealed, which have guiding significance for the study of the adsorption of different gases by anthracite. Moreover, the rationality of the model was proven from the perspective of the internal pore structure of the coal body. In addition, during the gas adsorption experiment on the coal conducted using the volumetric method, to obtain the pore volume inside the coal sample, the He gas was usually used for the testing. Since the kinetic radius of the He gas is 0.031 nm, which is less than those of $CO_2$, $CH_4$, and $N_2$, the pore volume that the He gas can detect and enter is greater than the pore volumes that $CO_2$, $CH_4$, and $N_2$ can detect and enter. This caused the amount of gas adsorption obtained in the experimental results to be lower than the true level. However, it can be calculated using the pore structure of the macromolecular model of coal. For example, the volume of smokeless micropores measured using the He gas in this study was 4487.71 $Å^3$/g, and the volume of micropores detected using $CH_4$ was 1867.55 $Å^3$/g, with a difference of 2620.16 $Å^3$/g. Therefore, this method can be used to assist the experiment and make up for the deficiency of the experiment, which can be used to improve the accuracy of the evaluation of coalbed methane reserves.

### 3.3. Analysis of the influence of temperature on coalbed methane adsorption

Temperature is one of the important factors affecting the adsorption of coalbed methane. In the process of coalbed methane development and $CO_2$ storage, the reservoir temperature is positively correlated with the burial depth of the reservoir. Therefore, it is very important to clarify the influence of the adsorption temperature on coal adsorption behavior for the scientific management of coalbed methane [35,36]. Given this, the $CO_2$, $CH_4$, and $N_2$ adsorption characteristics of the anthracite molecular model were simulated at different temperatures,

and the mechanism by which the temperature and gas properties influence the adsorption capacity was revealed from the microscopic point of view, laying a theoretical foundation for the molecular level study of the gas adsorption behavior of the coal macromolecular model. This provides technical support for the development of reasonable drainage methods for coalbed methane wells.

The adsorption kinetics curves for small $CO_2$ gas molecules on the anthracite macromolecular model at different temperatures are shown in Fig 8a. From the perspective of the adsorption capacity, within the simulated temperature range, the adsorption capacity of small $CO_2$ gas molecules on the anthracite macromolecular structure increased with increasing pressure at different temperatures, and the adsorption process could be divided into two stages: the initial fast adsorption stage (P < 3 MPa) and the gentle adsorption stage (P > 3 MPa). The Langmuir model was used to fit the adsorption isotherms of the $CO_2$ gas on coal, and the adjusted $R^2$ (Adj.$R^2$) value was greater than 0.99, indicating that the Langmuir model was suitable for the adsorption of $CO_2$ gas molecules by coal macromolecular models at different temperatures. With increasing temperature, the Langmuir adsorption constant of $CO_2$ adsorbed by the anthracite macromolecular model decreased gradually. With increasing temperature (293.15–323.15K), the saturated adsorption capacity of anthracite decreased by 22.46%, indicating that the increase in temperature had the strongest inhibitory effect on the adsorption of $CO_2$ by anthracite. This also revealed that the adsorption of $CO_2$ by anthracite was an exothermic reaction. To further describe the influence of temperature on the adsorption of $CO_2$ gas molecules by the anthracite macromolecular model, the curves of the adsorption increment of the anthracite macromolecular model with gas pressure were drawn and fitted (Fig 8d). From the perspective of the gas adsorption increment, the adsorption increment of small $CO_2$ gas molecules decreased with increasing pressure for the anthracite macromolecular model at all of the studied temperatures, and the fitting curve conforms to exponential attenuation. This is because the energy distribution on the surface of the coal macromolecular structure was uneven. At the beginning of adsorption, the lower the temperature was, the smaller the activation energy of gas molecules was, and more adsorption sites could be occupied on the surface of the coal. However, the number of adsorption sites on the surface of coal molecules was effective. As the adsorption behavior progressed, the number of adsorption sites on the coal decreased, and it tended to become saturated. With increasing adsorption temperature, the attenuation coefficient gradually decreased; that is, the attenuation rate of the $CO_2$ adsorption increment was inhibited at high temperatures. When the pressure was > 3 MPa, the adsorption increment change rate curve of the anthracite coal sample at any temperature was coincident, i.e., close to 0. This indicates that the temperature had an obvious effect on the change in the adsorption increment in the rapid adsorption stage, and the effect on the change in the adsorption increment was weak in the later stage of adsorption.

To better understand the adsorption difference of $CO_2$ on the anthracite macromolecular model at different temperatures from the microscopic level, the adsorption conformation diagram of $CO_2$ molecules on the anthracite single molecular structure at different temperatures (pressure 5 MPa) was obtained via molecular simulation (Fig 9a). In the same coal single molecule model, with increasing temperature, the number of $CO_2$ molecules adsorbed gradually decreased. As the temperature increased from 293.15 K to 323.15 K, the number of $CO_2$ molecules adsorbed by the anthracite single molecule model decreased from 15 to 11. From the microscopic point of view, the difference in the $CO_2$ adsorption behavior of anthracite at different temperatures was explained.

The adsorption isotherms of $CH_4$ gas molecules on the anthracite macromolecular model at different temperatures are shown in Fig 8b. The adsorption law of $CH_4$ on the anthracite macromolecular model is similar to that of $CO_2$. At the same temperature and pressure, the

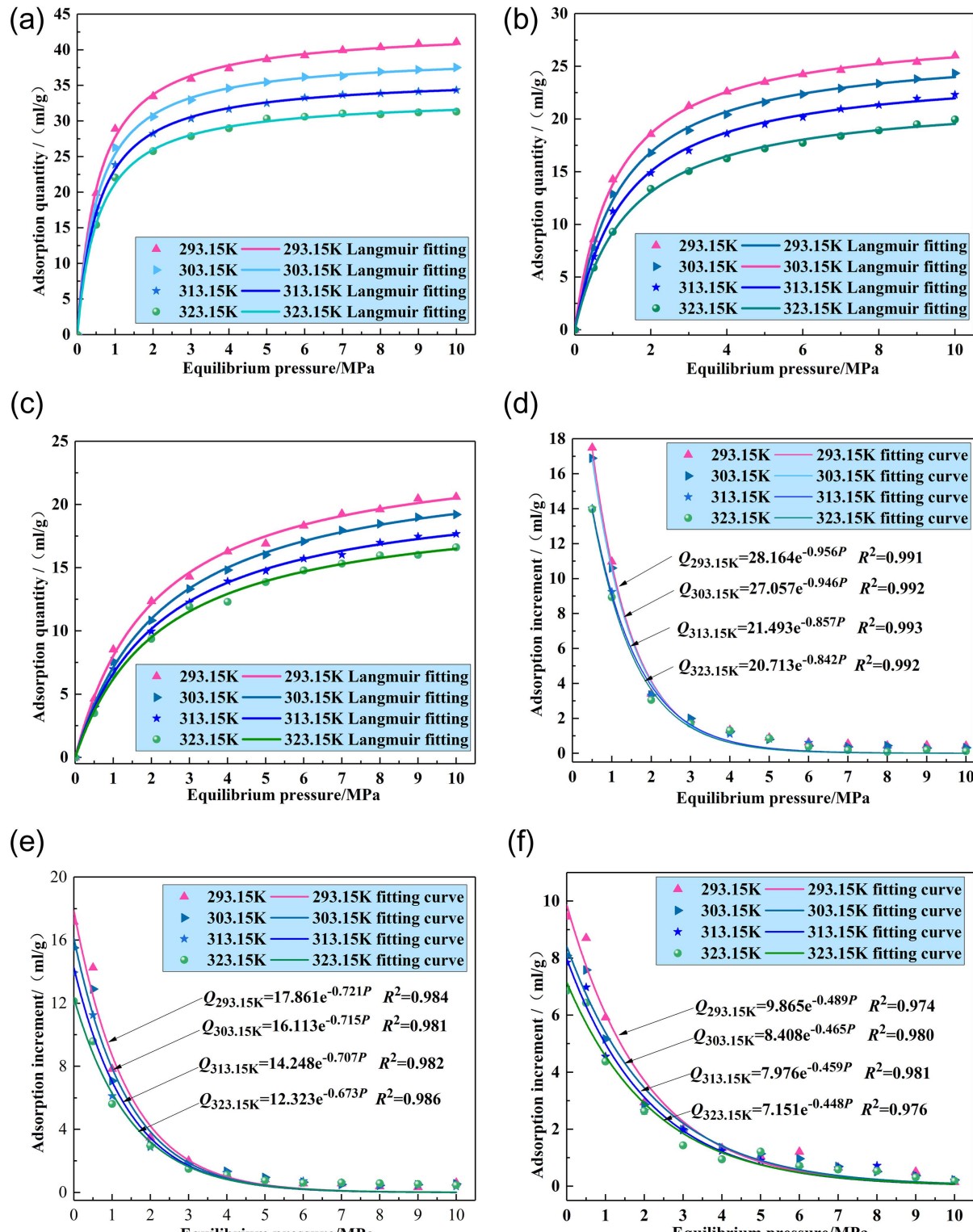

**Fig 8. Adsorption kinetics curves of CH$_4$, CO$_2$, and N$_2$ on the anthracite macromolecular model at different temperatures.** (a) CO$_2$ adsorption quantity (b) CH$_4$ adsorption quantity (c) N$_2$ adsorption quantity (d) CO$_2$ adsorption increment (e) CH$_4$ adsorption increment (f) N$_2$ adsorption increment.

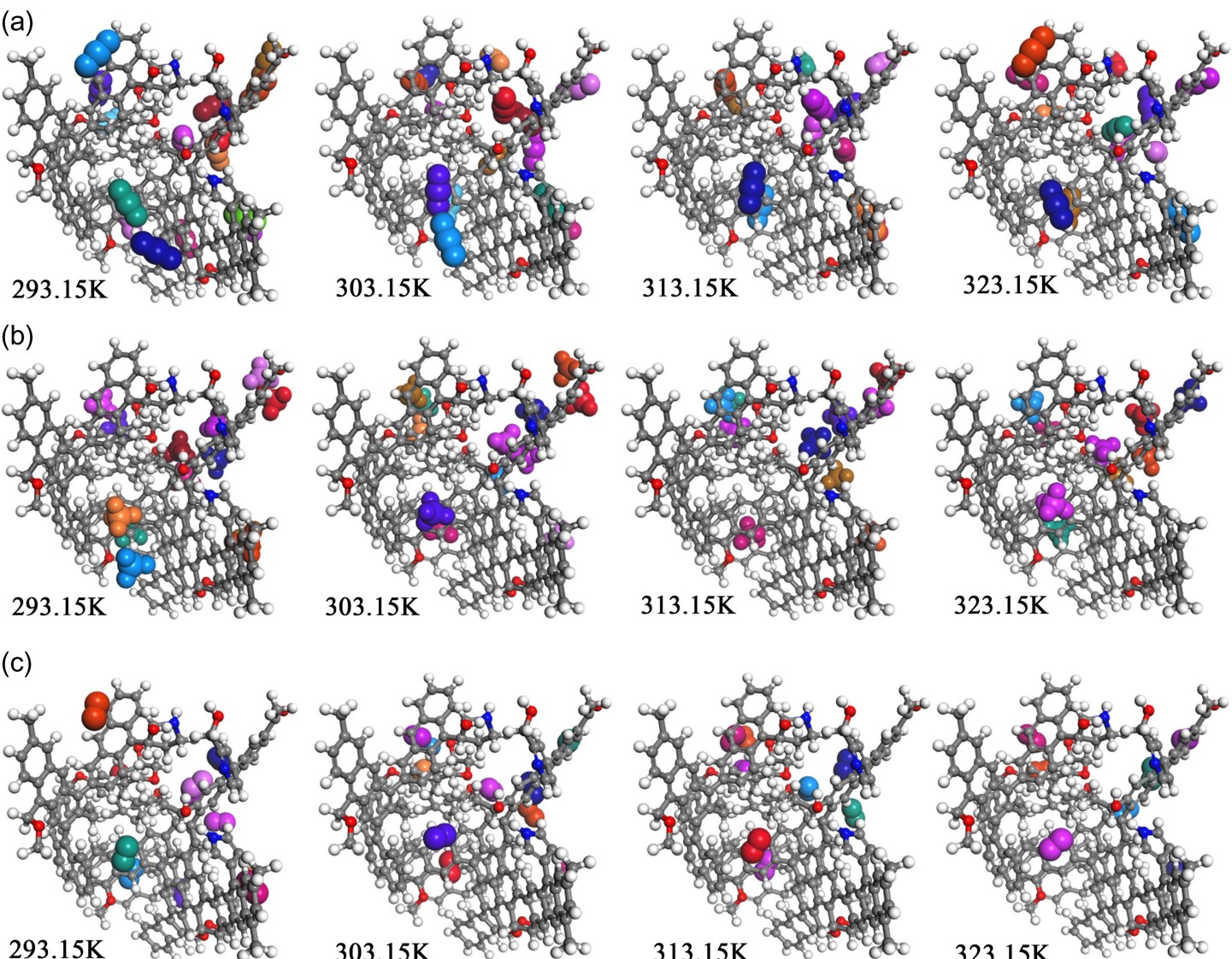

**Fig 9. $CO_2$, $CH_4$, and $N_2$ adsorption conformations of the anthracite monomolecular structure at different temperatures.** (a) $CO_2$ adsorption conformation (b) $CH_4$ adsorption conformation (c) $N_2$ adsorption conformation.

$CH_4$ adsorption capacity of the anthracite macromolecular model is less than the $CO_2$ adsorption capacity. This is mainly because the internal functional groups and pore structure of the coal macromolecular model have different adsorption capacities for the two gases, and they are closely related to the kinetic radius, critical temperature, molecular polarity, and diffusivity of the two gases. The curve of the adsorption increment of the anthracite macromolecular model with gas pressure was drawn and fitted (Fig 8e). With increasing adsorption temperature, the amount of $CH_4$ adsorption on the macromolecular model of anthracite decreased, and the Langmuir adsorption constant also exhibited similar characteristics. Within the simulated pressure range, a high temperature was not conducive to the adsorption of $CH_4$ on the macromolecular model of anthracite, indicating that the adsorption of $CH_4$ gas molecules on the coal surface was an exothermic process. Among them, as the temperature increased from 293.15 K to 323.15 K, the amount of $CH_4$ saturated adsorption on the anthracite decreased by

20.48%, indicating that the temperature change had the most significant effect on the amount of $CH_4$ adsorption on the anthracite.

To better understand the differences in the adsorption of $CH_4$ gas molecules by the anthracite macromolecular structure model at different temperatures at the microscopic level, the anthracite macromolecular model was selected under the condition of an adsorption pressure of 5 MPa. The adsorption conformations of $CH_4$ at different temperatures were obtained via molecular simulation (Fig 9b). With increasing temperature, for the same coal molecular model, the number of single $CH_4$ molecules adsorbed gradually decreased. Between 293.15 K and 323.15 K, the number of molecules adsorbed by anthracite decreased from 13 to 9, which explains the difference in the adsorption of $CH_4$ by anthracite at different temperatures from the molecular level.

In addition to $CO_2$ gas injection, $N_2$ injection can also improve the recovery rate of $CH_4$ when gas injection is used to enhance $CH_4$ extraction. To better clarify the effect of $N_2$ injection on enhanced $CH_4$ extraction, the adsorption kinetics of $N_2$ on the molecular structure of anthracite were studied. The adsorption isotherm curve of $N_2$ for the anthracite molecular model at different temperatures is shown in Fig 8c. With increasing temperature, the adsorption capacity and adsorption constant of $N_2$ on the anthracite macromolecular model decreased, indicating that a high temperature was not conducive to the adsorption of $N_2$ on the coal macromolecular model, which was similar to the adsorption results for $CH_4$ and $CO_2$. With increasing temperature, the saturated adsorption capacity of anthracite decreased by 18.42%, indicating that the increase in temperature had a significant inhibitory effect on the adsorption of $N_2$ by anthracite. The curve of the incremental change rate of $N_2$ adsorption on the anthracite macromolecular model was drawn and fitted (Fig 8f). It can be seen that as the temperature increased, the attenuation coefficient of the fitting curve gradually decreased. When anthracite adsorbed $N_2$ at any temperature, when P > 5 MPa, the adsorption incremental change curve coincided, indicating that the effect of temperature on the adsorption incremental change of $N_2$ was larger than that on the adsorption of $CH_4$ and $CO_2$.

To more clearly show the adsorption difference of $N_2$ on anthracite at different temperatures, the adsorption conformation of $N_2$ on the anthracite single molecule model at 5 MPa is shown in Fig 9c. With increasing temperature, the number of $N_2$ molecules adsorbed gradually decreased. As the temperature increased from 293.15 K to 323.15 K, the number of $N_2$ molecules adsorbed by anthracite decreased from 11 to 7, which explains the difference in the $N_2$ adsorption by anthracite at different temperatures from the microscopic perspective.

In summary, through analysis of the coalbed methane adsorption capacity of anthracite at different temperatures, it was found that at any temperature, the adsorption capacity of small $CO_2$, $CH_4$, and $N_2$ gas molecules on the anthracite model increases with increasing adsorption pressure, while the gas adsorption increment of the coal exhibits the opposite trend. Among them, the order of adsorption capacity of the three gases is $CO_2 > CH_4 > N_2$, which is consistent with the law obtained in this experimental test. which is consistent with the law obtained from the experiments conducted in this study. An increase in temperature is not conducive to the adsorption of gas molecules by the coal macromolecular model. The reason for this is that an increase in the adsorption temperature can promote increases in the energy, activity, and kinetic energy of gas molecules, which are not conducive to the capture of gas molecules on the surface of coal molecules during the adsorption process. Furthermore, a high temperature will inhibit the transformation of gas molecules from the free state to the adsorbed state. Some stable adsorbed gas molecules will also be desorbed and converted into active free gas due to the high temperature, so the gas adsorption capacity of coal will decrease with increasing temperature.

## 4. Conclusions

In this study, the characteristics of $CO_2$, $CH_4$, and $N_2$ adsorption on columnar coal were studied using an experimental platform for coal rock gas adsorption tests under multi-physical field conditions. Compared with the use of pulverized coal as the research object, the experimental arrangement is closer to the working condition of coal seam, which improves the accuracy of the conclusion. Moreover, the molecular simulation method is used to produce differences in the pore information obtained by different gases as the probe, which supports the experimental results. The simulation results can supplement the shortcomings of the experiment in observing microscopic changes. The main conclusions of this study are as follows.

(1) The gas adsorption capacities of anthracite are $CO_2 > CH_4 > N_2$, and the adsorption capacity increases with increasing gas injection pressure. When the pressure is less than 3 MPa, the adsorption capacities of anthracite for the three gases increase relatively quickly, and the isothermal adsorption curves are steep. When the pressure is greater than 3 MPa, the adsorption capacities of anthracite for $CH_4$ and $N_2$ increase relatively slowly, and the adsorption capacities tend to become saturated, while the $CO_2$ adsorption capacity exhibits a steep increase at approximately 4 MPa.

(2) The distribution characteristics of the micropores and pores in the macromolecular structure model of anthracite were revealed using the probe method. The smaller the kinetic diameter of gas molecules is, the more pores the gas can enter, and the larger the pore surface area that can come in contact with is. There are both accessible and inaccessible micropores.

(3) The $CO_2$, $CH_4$, and $N_2$ adsorption capacities of the anthracite model increased with increasing adsorption pressure, while the gas adsorption increment of the coal exhibited the opposite trend, which is consistent with the law obtained from the experiments.

(4) The $CO_2$, $CH_4$, and $N_2$ adsorption conformation models of the anthracite monomolecular structure at different temperatures were established. An increase in temperature was not conducive to the adsorption of gas molecules by the coal macromolecular model, and the increase in temperature had the greatest influence on the adsorption capacity of $CO_2$, followed by those of $CH_4$ and $N_2$.

The methodology employed in this study can be applied to the investigation of the adsorption characteristics of other coal types, thereby offering a novel perspective on the adsorption characteristics of coal in diverse environmental contexts. The findings offer a molecular-level technical foundation for the design of industrial processes for methane recovery and $CO_2$ storage. It provides a theoretical foundation for the sustainable development of energy and the environment.

## Supporting information

**S1 Table. Physicochemical properties of $CO_2$, $CH_4$ and $N_2$.**
(DOCX)

**S1 Data. Minimal data set.**
(ZIP)

## Author contributions

**Conceptualization:** Dan Zhao.

**Data curation:** Dan Zhao.

**Formal analysis:** Dan Zhao.

**Funding acquisition:** Dan Zhao.

**Investigation:** Dan Zhao, Mincong Huang, Mingyu Tong.

**Methodology:** Dan Zhao, Baihong Chen.

**Software:** Dan Zhao, Mincong Huang, Qu Du.

**Validation:** Dan Zhao, Wenchang He.

**Visualization:** Dan Zhao, Xiaofei Ke.

**Writing – original draft:** Dan Zhao.

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
