## [Decision Letter · Decision Letter 0]

1 May 2024

PONE-D-24-04323The study on the adsorption characteristics of anthracite under different temperature and pressure conditionsPLOS ONE

Dear Dr. Zhao,

Thank you for submitting your manuscript to PLOS ONE. After careful consideration, we feel that it has merit but does not fully meet PLOS ONE’s publication criteria as it currently stands. Therefore, we invite you to submit a revised version of the manuscript that addresses the points raised during the review process.

We look forward to receiving your revised manuscript.

Kind regards,

Afzal Husain Khan

Academic Editor

PLOS ONE

Journal Requirements:

2. Thank you for stating the following financial disclosure: "This work was partly supported by the Scientific Research Project of Guangdong Provincial Department of Education—Young Innovative Talents Project（grant number 2022KQNCX141".  

3. Thank you for stating the following in the Acknowledgments Section of your manuscript: "This work was partly supported by the Scientific Research Project of Guangdong Provincial Department of Education—Young Innovative Talents Project（grant number 2022KQNCX141）."

Please remove any funding-related text from the manuscript and let us know how you would like to update your Funding Statement. Currently, your Funding Statement reads as follows: "This work was partly supported by the Scientific Research Project of Guangdong Provincial Department of Education—Young Innovative Talents Project（grant number 2022KQNCX141".  

Reviewers' comments:

Reviewer's Responses to Questions

**Comments to the Author**

1. Is the manuscript technically sound, and do the data support the conclusions?

Reviewer #1: Yes

Reviewer #2: Partly

2. Has the statistical analysis been performed appropriately and rigorously? 

Reviewer #1: Yes

Reviewer #2: Yes

3. Have the authors made all data underlying the findings in their manuscript fully available?

Reviewer #1: Yes

Reviewer #2: Yes

4. Is the manuscript presented in an intelligible fashion and written in standard English?

Reviewer #1: Yes

Reviewer #2: Yes

5. Review Comments to the Author

Reviewer #1: The authors have conducted a moderate thorough literature review, undertaken a piece of data collection and have analyze information moderately. The Authors need to improve the abstract and introduction part of the part. Besides this the novelty of the work needs to be addressed in the paper. The paper also needs thorough English correction. The presentation of data is upto the mark.

Reviewer #2: In this paper, the effects of temperature and pressure on the adsorption of CO2, CH4 and N2 by anthracite were discussed by experiment and molecular dynamics simulation, which improved the understanding of the adsorption characteristics and mechanism of anthracite. The results provide theoretical support and technical support for coalbed methane control and CO2 geological storage. Here are some of my comments:

1. The figure is somewhat blurred, it is recommended to replace the high-definition figure.

2. The introduction needs to describe in more detail the background of the study, why anthracite coal was chosen as the subject of the study, and the specific implications of this study for coalbed methane control and CO2 geological storage. Some of the references have little to do with the research content of this paper, and it is suggested that appropriate deletions be made or more closely related articles be added.

3. Much has been done by previous authors to address the effect of temperature and pressure on the adsorption characteristics of anthracite. How to show the innovation of this paper?

4. Was the model validated for reasonableness?

5. The description of the molecular simulation section is too simple and needs to be supplemented with some information on simulation steps, parameter settings, and so on.

6. Section 3.2 Second paragraph units note superscript.

7. The conclusion section is too brief and it is recommended to summarize the main innovations, findings and implications of the article and point out directions for future work.

8. Harmonize reference formats as required.

6. PLOS authors have the option to publish the peer review history of their article (what does this mean? ). If published, this will include your full peer review and any attached files.

**Do you want your identity to be public for this peer review?** For information about this choice, including consent withdrawal, please see our Privacy Policy .

Reviewer #1: No

Reviewer #2: No

---

## [Author Response · Author response to Decision Letter 0]

3 May 2024

Dear editor:

Thank you very much for giving us an opportunity to revise our manuscript. We appreciate the editor and reviewers very much for their constructive comments and suggestions on our manuscript entitled“The study on the adsorption characteristics of anthracite under different temperature and pressure conditions”(ID: PONE-D-24-04323).

We have studied reviewers' comments carefully. According to the reviewers' detailed suggestions. we have made a careful revision on the original manuscript. I modified the copy of the markup to form a separate file, named 'Revised Manuscript with Track Changes'.

Kind regards.

Corresponding author: Dan Zhao

E-mail address: 2373562966@com

Replies to the reviewers’ comments:

Reviewer #1:

1. In the introduction section what does the Author mean by “stress condition” Please explain.

Response: I am very sorry for your misunderstanding due to my unclear expression. "stress condition" in the article refers to the triaxial stress exerted on coal pillar during the experiment, and "stress condition" can also be understood as a set of variables set in the experiment.

2. I suggest to further elaborate more about point of zero charge analysis to know the surface charge and its effect should be included in the methodology and results section. The author can refer these articles https://www.tandfonline.com/doi/full/10.1080/07391102.2023.2186704.

Response: Thanks for the reviewer's suggestions, I have read this article closely and found that this article describes the surface charge and its influence law in detail, which has a strong guidance and reference significance for the molecular simulation part of this paper, so I cite this article, which is reflected as follows:

[34]Çetinkaya Serap,Eyupoglu Volkan,Çetintaş Halil İbrahim, et al. Removal of Erythrosine B dye from wastewater by Penicillium italicum: experimental, DFT, and molecular docking studies.[J].Journal of biomolecular structure dynamics,2023,41(23):11-12. doi: 10.1080/07391102.2023.2186704.

3. What is the novelty of this work? The author should add more lines related to it in the introduction section or below the abstract.

Response: Thanks for the reviewer's suggestions on innovation points. Due to the problems I expressed, the innovation points in this paper are too scattered and there is no clear condensed innovation points. The innovation of this paper is mainly divided into two points. The first innovation is that the previous authors mostly used pulverized coal for experimental research. In this paper, cylindrical raw coal was used for experiments, and based on the coal-rock adsorption test platform under multi-physical field conditions, the isothermal adsorption experiment of CO2/CH4/N2 by cylindrical anthracite was carried out. The experimental arrangement is closer to the working condition of the coal seam, and the law of the change of adsorption capacity with gas injection pressure is obtained. The second innovation is the establishment of the CH4/CO2/N2 adsorption conformation model of single molecule structure of anthracite at different temperatures, and the pore information obtained by different gases as probes is different, and the simulation results can supplement the deficiency that the experiment cannot observe the microscopic changes. Based on this, I made supplementary revisions in the introduction part of the manuscript, as follows:

Their results revealed that the model can predict the adsorption behavior of CH4, N2, and CO2 on coal and considers the influence of adsorption surface structure characteristics on the adsorption behavior, which can improve the ability to predict high-pressure adsorption phenomenon. Through the above analysis, we can see that at present, some research has been conducted [22–24]. However, there are fewer experiments that use columnar coal for analysis, and there is relatively little research on the adsorption micro-mechanism using molecular simulation combined with experiments.

Given this, in this paper, a coal rock adsorption test platform under multi-physical field conditions is applied to explore the variations in and controlling mechanism of the coal adsorption capacity with adsorption pressure during the adsorption of CO2, CH4, and N2 by columnar anthracite from the Yangquan Coal Mine, Shanxi Province, China, under stress conditions. Based on experimental research and through molecular dynamics simulation, the pore structure characteristics of an anthracite macromolecular structure model and the influence of temperature on the adsorption performance of the anthracite molecular structure model for single component CO2, CH4, and N2 gases were studied from the microscopic point of view. The research findings presented in this paper are valuable for enhancing our understanding of the adsorption characteristics and mechanisms of CH4, CO2, and N2 in coal. The aim is to elucidate the influence of factors such as adsorption pressure, temperature, gas properties, and other variables on the molecular-level adsorption and diffusion behaviors of gases. This study provides theoretical support and technical guidance for optimizing CH4 treatment and CO2 geological storage in coal mines.

4. Please mention properly about the procedures of recyclability of the adsorbents used in this study as it reflects the sustainability and cost-effectiveness of the adsorbents e.g. see”. Please explain. (Cite these papers as an example.) https://link.springer.com/article/10.1007/s13369-022-07015-w, https://www.sciencedirect.com/science/article/pii/S0927775721017489.

Response: Thanks for the reviewer's suggestions, I read this article closely and found that these two articles described the recyclability procedure of adsorbent in detail, reflecting the sustainability and cost-effectiveness of adsorbent, which provided strong guidance and reference significance for the research on coal adsorption in this paper. Therefore, these two articles were quoted and reflected in the introduction of the article as follows:

[13]Tauqir A ,Saood M M ,Ullah S K , et al.Synthesis and Adsorptive Performance of a Novel Triazine Core-Containing Resin for the Ultrahigh Removal of Malachite Green from Water[J].Arabian Journal for Science and Engineering,2022,48(7):8571-8584. doi：10.1007/S13369-022-07015-W.

[15]Dalhat N M ,Saood M M ,Mukarram Z , et al.Volcanic ashe and its NaOH modified adsorbent for superb cationic dye uptake from water: Statistical evaluation, optimization, and mechanistic studies[J].Colloids and Surfaces A: Physicochemical and Engineering Aspects,2022,634. doi：10.1016/J.COLSURFA.2021.127879.

Thanks to the reviewer for carefully reading my manuscript. According to your suggestions, I read my manuscript again and corrected some language errors. Your comments on the review of this paper have improved the academic level of this paper as a whole, and I would like to express my thanks to you again!

Reviewer #2:

1. The figure is somewhat blurred, it is recommended to replace the high-definition figure. 1.

Response: Thanks for the reviewer's careful review of my manuscript. Due to my negligence, some numerical simulation images were not clear, so I revised and uploaded the high-definition figures again. The revised figures are as follows:

(a) Three-dimensional model of anthracite (b) Crystal cell model

Fig.2 Numerical model of anthracite

(a) CO2 gas (b) CH4 gas

(c) N2 gas

Fig.4 Variation in the gas adsorbed by the coal samples under effective stress

Fig. 6 Position of each slice of anthracite

2. The introduction needs to describe in more detail the background of the study, why anthracite coal was chosen as the subject of the study, and the specific implications of this study for coalbed methane control and CO2 geological storage. Some of the references have little to do with the research content of this paper, and it is suggested that appropriate deletions be made or more closely related articles be added.

Response: Thanks to the reviewer for pointing out the shortcomings of the introduction part of this paper. I fully agree with your point of view and greatly enhance the theoretical value of this paper through your guidance. First, I added the main reasons for choosing anthracite as my research topic in the introduction, and explained the specific impact of this research on coal-bed methane control and CO2 geological storage. Some less relevant references are deleted, and more closely related references about coal bed methane are added. Specific modifications are as follows:

As the main energy source in China, coal has provided strong support for social and economic development and will continue in its role as a ballast stone in maintaining stable economic operations for a long time in the future [1,2]. China's anthracite resources are very rich, accounting for 14% of the country's total coal reserves, while anthracite contains a large number of coal-bed methane resources, but coal mine gas disasters still occur from time to time [3,4]. The main reason for their occurrence is that the research on the development theory and technology of coalbed methane (CH4, CO2, N2, and other gases) is relatively weak. In particular, the understanding of the adsorption mechanism of coalbed methane is still insufficient, resulting in the unsatisfactory development effect of coalbed methane. Therefore, it is urgent to study the adsorption characteristics of coalbed methane in a reservoir [5]

Given this, in this paper, a coal rock adsorption test platform under multi-physical field conditions is applied to explore the variations in and controlling mechanism of the coal adsorption capacity with adsorption pressure during the adsorption of CO2, CH4, and N2 by columnar anthracite from the Yangquan Coal Mine, Shanxi Province, China, under stress conditions. Based on experimental research and through molecular dynamics simulation, the pore structure characteristics of an anthracite macromolecular structure model and the influence of temperature on the adsorption performance of the anthracite molecular structure model for single component CO2, CH4, and N2 gases were studied from the microscopic point of view. The research findings presented in this paper are valuable for enhancing our understanding of the adsorption characteristics and mechanisms of CH4, CO2, and N2 in coal. The aim is to elucidate the influence of factors such as adsorption pressure, temperature, gas properties, and other variables on the molecular-level adsorption and diffusion behaviors of gases. This study provides theoretical support and technical guidance for optimizing CH4 treatment and CO2 geological storage in coal mines.

References [9] and [17] in the original manuscript have been deleted, and new references related to coal bed methane have been added, as follows:

Ranathunga[9] studied the applicability of CO2 to enhance coalbed methane recovery through CO2-driven coalbed methane experiment, and the results showed that CO2 injection had a higher coalbed methane recovery rate than natural extraction, and effectively improved the anti-reflection performance of coal seams.

Zeng[17] simulated the coal-bed methane mining process using the adsorption-strain coupling model, and the results showed that the microsimulation could explore the changes in coal deformation and permeability that were difficult to find in experiments.

3. Much has been done by previous authors to address the effect of temperature and pressure on the adsorption characteristics of anthracite. How to show the innovation of this paper?

Response: Thanks for the reviewer's suggestions on innovation points. Due to the problems I expressed, the innovation points in this paper are too scattered and there is no clear condensed innovation points. The innovation of this paper is mainly divided into two points. The first innovation is that the previous authors mostly used pulverized coal for experimental research. In this paper, cylindrical raw coal was used for experiments, and based on the coal-rock adsorption test platform under multi-physical field conditions, the isothermal adsorption experiment of CO2/CH4/N2 by cylindrical anthracite was carried out. The experimental arrangement is closer to the working condition of the coal seam, and the law of the change of adsorption capacity with gas injection pressure is obtained. The second innovation is the establishment of the CH4/CO2/N2 adsorption conformation model of single molecule structure of anthracite at different temperatures, and the pore information obtained by different gases as probes is different, and the simulation results can supplement the deficiency that the experiment cannot observe the microscopic changes.

4. Was the model validated for reasonableness?

Response: Thanks for the reviewer's questions about the verification of the model's rationality. I strongly agree with the reviewer's suggestions, because the rationality of the model structure directly affects the accuracy of the simulation results, so I am very sorry for causing your misunderstanding due to the problems I expressed. Since the model I cited was self-constructed and the model construction process and rationality verification have been published in the early edition of Plos one, I did not give too much description. After referring to your opinions, I added some general descriptions about this part in the paper, and your suggestions made the expression of this paper more clear.

The references for the anthracite model I independently constructed are as follows:

[26]Jia Jinzhang,Wu Yumo,Zhao Dan, et al. Molecular structure characterization analysis and molecular model construction of anthracite.[J]. PloS one,2022,17(9). doi:10.1371/JOURNAL.PONE.0275108.

In the article on the construction of anthracite model, the expression on the rationality verification of the model is as follows:

4.2 The anthracite coal molecular model validation

To further verify the rationality of the parameters of the molecular model of anthracite, the molecular structure cell model of anthracite was established, and the adsorption of CH4 gas molecules in the molecular structure model of anthracite was observed. The molecular structure models of 15 anthracites were obtained, and the Amorphous Cell module was used. The calculation accuracy was fine, and we used the COMPASS force field. The 15 single molecular structures were placed into the cell to add three-dimensional periodic boundary conditions, and the density was set to 1.32 g/cm3. Then, structure optimization and dynamic optimization of the cell model of anthracite were carried out to minimize and stabilize the energy of the constructed coal molecular structure model. Finally, the structure model size of anthracite with low energy conformation was A = B = C = 3.89034 nm, where the molecular formula was C3120H2430N60O180. The cell model of the anthracite macromolecular structure is shown in Fig. 12.

Fig 12 Cellular model of the molecular structure of anthracite

MS software was used to analyze the adsorption of CH4 gas molecules in the molecular structure model of anthracite by combining GCMC and MD. The simulation process was completed by the adsorption and Forcite modules in MS. The adsorption data of the CH4 molecules in the anthracite molecular model were compared with the experimental data of anthracite in reference 17, as shown in Fig. 13. The comparison showed that the adsorption amount of CH4 gas was within an order of magnitude, which proved that the anthracite molecular model established in this work conformed to the structural characteristics of anthracite. However, the columnar coal samples used in the adsorption experiment had defects to some extent, resulting in a decrease in the pore volume and specific surface area in the coal body, which reduced the gas adsorption capacity in the experimental process. The variation trend of the CH4 adsorption amount obtained by the experiments and molecular simulation was basically consistent, which further proved that the anthracite molecular model established in this work was reasonable.

Fig 13 Comparison of CH4 adsorption simulation and experiment

5. The description of the molecular simulation section is too s

---

## [Decision Letter · Decision Letter 1]

29 Aug 2024

PONE-D-24-04323R1The study on the adsorption characteristics of anthracite under different temperature and pressure conditionsPLOS ONE

Dear Dr. Zhao,

Thank you for submitting your manuscript to PLOS ONE. After careful consideration, we feel that it has merit but does not fully meet PLOS ONE’s publication criteria as it currently stands. Therefore, we invite you to submit a revised version of the manuscript that addresses the points raised during the review process.

Authors should submit all original source materials (including raw data from the characterization center in an Excel format, metadata etc., for experimental works while for modeling/simulation works raw data, model files, etc.) to the journal in the next submission as supporting information. Authors should declare in the cover letter that part or all of this manuscript is not generated by an AI generation tool (e.g., ChatGPT).

We look forward to receiving your revised manuscript.

Kind regards,

Mashallah Rezakazemi

Academic Editor

PLOS ONE

Reviewers' comments:

Reviewer's Responses to Questions

**Comments to the Author**

1. If the authors have adequately addressed your comments raised in a previous round of review and you feel that this manuscript is now acceptable for publication, you may indicate that here to bypass the “Comments to the Author” section, enter your conflict of interest statement in the “Confidential to Editor” section, and submit your "Accept" recommendation.

Reviewer #3: All comments have been addressed

Reviewer #4: (No Response)

Reviewer #5: All comments have been addressed

2. Is the manuscript technically sound, and do the data support the conclusions?

Reviewer #3: Yes

Reviewer #4: Partly

Reviewer #5: Yes

3. Has the statistical analysis been performed appropriately and rigorously? 

Reviewer #3: Yes

Reviewer #4: Yes

Reviewer #5: No

4. Have the authors made all data underlying the findings in their manuscript fully available?

Reviewer #3: Yes

Reviewer #4: Yes

Reviewer #5: Yes

5. Is the manuscript presented in an intelligible fashion and written in standard English?

Reviewer #3: Yes

Reviewer #4: Yes

Reviewer #5: Yes

6. Review Comments to the Author

Reviewer #3: The authors have improved the manuscript, and most of the comments have been taken into consideration. Consequently, I suggest publishing the paper after minor revision. It is worth to be noted that the unit of specific surface area and total pore volume are Å2/g and Å3/g, respectively. Please correct these units in the manuscript.

Reviewer #4: This work presented a study on the potential of anthracite coal as an effective medium for adsorbing CO2. Nonetheless, the language of this paper needs to be improved at the academic level as a whole. Active or passive sentences were not carefully described. The Authors must correct the language errors

Reviewer #5: The manuscript titled "The study on the adsorption characteristics of anthracite under different temperature and pressure conditions" offers valuable insights into how anthracite interacts with CO₂, CH₄, and N₂ under various temperature and pressure scenarios. Given the increasing focus on carbon sequestration and methane recovery, this research is both timely and relevant. However, after carefully reviewing the manuscript, I believe it needs significant revisions before it can be considered for publication. Below, I’ve outlined the main concerns and recommendations that should be addressed in the revised version.

Abstract:

1. The abstract effectively summarizes the key findings regarding the adsorption capacities of CO₂, CH₄, and N₂ on anthracite under varying conditions. However, it lacks a discussion on the practical implications of these findings. How do these results inform the potential for industrial applications, particularly in CO₂ sequestration or CH₄ recovery?

2. The abstract should also mention the methodology, particularly the use of molecular simulation alongside experimental approaches, as this is a significant aspect of the study.

3. Consider adding a brief mention of the study's limitations, such as the reliance on a specific coal type (anthracite) and how this might affect the generalizability of the findings.

4. The paragraph uses a repetitive sentence structure, with most sentences beginning in a similar way ("The study of...," "To explore...," "The results show..."). This can make the text monotonous and harder to engage with. Varying sentence structures would make the text more dynamic and easier to read.

5. The text could be more concise. Phrases like "in this paper, columnar anthracite is taken as the research object" could be simplified to "this study focuses on columnar anthracite.

Introduction:

1. The introduction provides a thorough background on the significance of studying adsorption characteristics in coal, particularly anthracite. However, it could be condensed to improve readability and focus more directly on the study's objectives.

2. While the introduction references prior studies, it could benefit from a more detailed comparison with previous research, especially in terms of how this study advances our understanding of the topic. Specifically, what gaps in the literature does this study aim to fill? To strengthen this section, I recommend citing recent studies that have explored similar topics in the context of carbon capture and storage (CCS) and methane recovery, such as those by https://doi.org/10.1016/j.envres.2023.116879 and https://doi.org/10.1016/j.jece.2023.110833. These articles provide relevant insights and comparisons that can help contextualize the novelty of your work, particularly in how your study contributes to the ongoing development of efficient adsorption technologies. Citing these sources will also demonstrate how your research builds upon and extends the current state of knowledge in this critical area.

3. The introduction should highlight the novelty of the study more clearly. What distinguishes this study from previous work on similar topics? Is it the combination of experimental and simulation approaches, the specific conditions tested, or the focus on anthracite?

4. While the text does a thorough job of summarizing past studies, it does not sufficiently highlight how this study contributes something new to the field. The discussion of previous research dominates the introduction, but the authors do not clearly articulate how their work differs from or builds upon these studies.

Materials and Methods:

1. The methodology is well-detailed, particularly the description of the experimental setup and the conditions under which the adsorption tests were conducted. However, there is limited discussion on the reproducibility of the experiments. How were variations in the coal samples (e.g., porosity, moisture content) controlled or accounted for?

2. The rationale behind the selection of the specific temperature and pressure ranges should be explained. How do these conditions relate to real-world scenarios in coal mining or carbon sequestration projects?

3. The molecular simulation method is described, but the choice of parameters (e.g., force field, probe radius) should be justified. Why were these particular settings chosen, and how do they affect the simulation outcomes?

Results and Discussion:

1. The results are presented clearly, with appropriate use of figures and tables to illustrate key findings. However, the discussion often remains descriptive, with limited interpretation of the results. For example, the observed trends in adsorption with varying pressure and temperature could be linked more explicitly to the molecular interactions at play.

2. The study uses the Langmuir model to fit the adsorption data, but the choice of this model over others (e.g., Freundlich, Temkin) is not justified. A discussion on the model's applicability to the experimental data, including any limitations or deviations, would strengthen the analysis.

3. The discussion on the impact of temperature on adsorption is thorough but could benefit from a comparison with similar studies. How do the findings compare with previous research on different coal types or adsorption conditions?

4. The molecular simulation results are interesting but could be integrated more closely with the experimental findings. For instance, how do the pore structures observed in the simulations correspond to the experimental adsorption data?

5. Please add error bars to all relevant figures to indicate data variability and improve result interpretation.

6. The manuscript focuses on anthracite, but the results would be more impactful if they were compared with data from other types of coal, such as bituminous or lignite. A brief comparative discussion would help place the findings in a broader context and highlight the specific advantages or disadvantages of using anthracite in adsorption processes.

Conclusion:

1. The conclusion effectively summarizes the key findings but could be expanded to include a discussion on the study's limitations and potential areas for future research. For example, how might the adsorption characteristics differ with other coal types or under different environmental conditions?

2. The conclusion should also address the practical implications of the study. How can these findings inform the design of industrial processes for methane recovery or CO₂ sequestration?

3. The final paragraph could benefit from a brief mention of the broader significance of the study in the context of energy and environmental sustainability.

7. PLOS authors have the option to publish the peer review history of their article (what does this mean? ). If published, this will include your full peer review and any attached files.

**Do you want your identity to be public for this peer review?** For information about this choice, including consent withdrawal, please see our Privacy Policy .

Reviewer #3: No

Reviewer #4: No

Reviewer #5: No

---

## [Author Response · Author response to Decision Letter 1]

2 Sep 2024

Dear editor:

Thank you very much for giving us an opportunity to revise our manuscript. We appreciate the editor and reviewers very much for their constructive comments and suggestions on our manuscript entitled“The study on the adsorption characteristics of anthracite under different temperature and pressure conditions”(ID: PONE-D-24-04323).

We have studied reviewers' comments carefully. According to the reviewers' detailed suggestions. we have made a careful revision on the original manuscript. I modified the copy of the markup to form a separate file, named 'Revised Manuscript with Track Changes'.

Kind regards.

Corresponding author: Dan Zhao

E-mail address: 2373562966@com

Replies to the reviewers’ comments:

Comments:

1. Both the abstract and conclusion parts need to show the innovation of this paper to differentiate the present work from others.

Response: Thank the reviewers for their suggestions on the abstract part of the manuscript. In order to highlight the innovation of this article, I revised the abstract as follows :

The study of the adsorption characteristics of coal is of great significance to gas prevention and CO2 geological storage. To explore the adsorption mechanism of coal, this study focuses on columnar anthracite. Adsorption tests on coal rock under a range of physical field conditions were conducted using the volumetric method. The adsorption characteristics of anthracite for CO₂, CH₄, and N₂ gases under different conditions were investigated using Grand Canonical Monte Carlo (GCMC) and Molecular Dynamics (MD) methods. The results showed that the adsorption capacities of anthracite for these three gases are in the order of CO2 > CH4 > N2, and that the adsorption capacity increases with increasing gas injection pressure. The CO2/CH4/N2 gas molecule adsorption capacity of the anthracite macromolecular structure model decreases with increasing temperature. The increase in temperature has the greatest influence on the CO2 absorption capacity, followed by the CH4 and N2 adsorption capacities. The research offers a theoretical basis for the control of coal mine gas and the geological storage of CO2.

2. What was the main reason for selecting the anthracite coal for this research? It should be mentioned carefully in detail at the beginning of the Introduction section.

Response: I would like to thank the reviewer for identifying the deficiencies in my Introduction section. I am deeply sorry that, due to my negligence, the primary reason for selecting anthracite as the subject of the experiment was not adequately emphasised. In light of the above, I have made the following revisions to the manuscript:

As a crucial raw material for deep coal processing, anthracite has been instrumental in the early stages of China's coal chemical industry. Concurrently, a considerable volume of coalbed methane resources is present within anthracite, yet incidents of coal mine gas disasters persist. The principal reason for this is that research into the development theory and technology of coalbed methane (CH4, CO2, N2 and other gases) is relatively limited. In particular, there is still a lack of understanding of the mechanism of adsorption characteristics of coalbed methane, which is resulting in an unsatisfactory development effect of coalbed methane. Accordingly, this study employs anthracite as the experimental coal sample to investigate the adsorption characteristics of coalbed methane in the reservoir, which represents a critical and pressing issue in the development and research of coalbed methane.

3. In section 3.1., it was mentioned that “The larger the polarizability of the gas molecule is, the greater the dispersion force and induction force are. This makes it easier for the gas to occupy the adsorption site on the surface of the”. It means that the higher polarizability of CO2 molecules facilitates their adsorption on the adsorbent’s surface compared to other gases. Therefore, it is recommended that the Authors work on the schematic diagram of interactions between anthracite and the desired molecule (i.e., CO2) for more clarification.

Response: Thank you for the reviewer 's conclusion that CO2 is more adsorbable than CH4 and N2 in a schematic way. I strongly agree with your opinion. I used molecular dynamics to simulate the adsorption configuration of three gases in seven main functional groups in coal. The conclusion is consistent with the experimental results, but this simulation study I and our team 's Professor Jinzhang J have published in the Fuel journal in 2023 ( Jinzhang J,Yumo W ,Dan Z , et al. Adsorption of CH4/CO2/N2 by different functional groups in coal[J].Fuel,2023,335 ), so this sentence I want to express is explained in the form of a conclusion. In this regard, I have added a reference to the article to illustrate this problem. The adsorption configurations of seven main functional groups of three gases in coal are as follows:

Stable configurations of CO2adsorbed by fragments of different functional groups

Stable configurations of CH4 adsorbed by fragments of different functional groups

Stable configurations of N2 adsorbed by fragments of different functional groups

4. What is the meaning of the “effective stress”? It has been mentioned several times in section 3.1 without any clarification. Please define it before and explain clearly how this factor(s) impacts the gas adsorption capacity of the anthracite used.

Response: I would like to thank the reviewer for the explanation of ‘effective stress’, and I am very sorry that I did not write down the origin of ‘effective stress’ in the paper due to my negligence. The ‘effective stress’ is the stress applied to the coal samples by the experimental test rig in this paper, which is based on the applied axial and peripheral pressures to simulate the depth of burial, reservoir pressure, temperature and geostress distribution of the coal samples. Therefore, I have added a description of the ‘effective stress’ setting in Summary 2.1 of the manuscript, as shown below:

In this experiment, the experimental test of adsorption of pure CH4, CO2 and N2 by anthracite coal was carried out by the isovolumetric method.According to the burial depth of the experimental coal samples, the reservoir pressure, the temperature and the distribution of the geostresses, the final setup of the experimental temperature was 30°C. The axial pressure of the test samples was applied at 10 MPa, the enclosing pressure was applied at 8 MPa, and the equilibrium pressure of the adsorption was taken as 7 points, ranging from 0.5 to 6 MPa.For the convenience of the calculations, the effective stress is defined as: 1 / 3 × [ Axial pressure + 2 × Confining pressure ] -Equilibrium pressure.

5. The language of this paper needs to be improved at the academic level as a whole. Active or passive sentences were not carefully described. The Authors must correct the language errors. Some are:

i. Abstract “The results show that the adsorption capacities of anthracite for these three gases are”. The results showed that...

ii. Section 1 Introduction “He gas had a weak adsorption capacity with an almost negligible volume change”. He is a specific gas. Use an article before that “The He”. It has been repeated many times in the drafts, please correct all.

iii. Section 1 Introduction “However, there are fewer experiments that use columnar coal for analysis, and there is relatively little research on the adsorption micro-mechanism using molecular simulation combined with experiments”. There are fewer experiments that should be changed to “fewer experiments use”.

iv. Section 2.1 “F8 were opened to vacuum the system to discharge any impurities such as air and water from the enture device”. What is the meaning of enture? It might be entire.

v. Section 2.1 “After the free volume test was completed, the system was vacuumized again to ensure that He gas inside the system and the coal sample was completely pumped out to avoid experimental errors”. There is no vocabulary in English for vacuumized. Vacuumed is the correct form.

vi. Section 3.2 “The smaller the radius of the probe molecule is, the larger the pore volume that can be detected is”. The second is should be deleted.

vii. Section 3.3 “To further explore the pore distribution characteristics of the anthracite macromolecular structure, on the basis of obtaining the pore distribution characteristics”. On the basis of should be changed to based on.

Response: Thank you for reading my manuscript carefully. I apologize for the numerous grammatical issues found in the manuscript and greatly appreciate your patience in helping me correct them. I have revised the entire text according to your suggested changes and have reviewed and amended the manuscript thoroughly using your method.

Reviewer #3: The authors have improved the manuscript, and most of the comments have been taken into consideration. Consequently, I suggest publishing the paper after minor revision. It is worth to be noted that the unit of specific surface area and total pore volume are Å2/g and Å3/g, respectively. Please correct these units in the manuscript.

Response: I would like to thank the reviewers for their valuable comments on my manuscript and their recognition of its quality. I apologize for any mistakes that were made due to my negligence, and I have made the necessary changes to the manuscript according to your suggestions.

Reviewer #4: This work presented a study on the potential of anthracite coal as an effective medium for adsorbing CO2. Nonetheless, the language of this paper needs to be improved at the academic level as a whole. Active or passive sentences were not carefully described. The Authors must correct the language errors

Response:I would like to thank the reviewers for their valuable comments on my manuscript and their recognition of its quality. I apologize for the grammatical errors in the manuscript due to my oversight. I have reviewed and revised the entire manuscript according to your suggestions.

Reviewer #5: The manuscript titled "The study on the adsorption characteristics of anthracite under different temperature and pressure conditions" offers valuable insights into how anthracite interacts with CO₂, CH₄, and N₂ under various temperature and pressure scenarios. Given the increasing focus on carbon sequestration and methane recovery, this research is both timely and relevant. However, after carefully reviewing the manuscript, I believe it needs significant revisions before it can be considered for publication. Below, I’ve outlined the main concerns and recommendations that should be addressed in the revised version.

Abstract:

1. The abstract effectively summarizes the key findings regarding the adsorption capacities of CO₂, CH₄, and N₂ on anthracite under varying conditions. However, it lacks a discussion on the practical implications of these findings. How do these results inform the potential for industrial applications, particularly in CO₂ sequestration or CH₄ recovery?

2. The abstract should also mention the methodology, particularly the use of molecular simulation alongside experimental approaches, as this is a significant aspect of the study.

3. Consider adding a brief mention of the study's limitations, such as the reliance on a specific coal type (anthracite) and how this might affect the generalizability of the findings.

4. The paragraph uses a repetitive sentence structure, with most sentences beginning in a similar way ("The study of...," "To explore...," "The results show..."). This can make the text monotonous and harder to engage with. Varying sentence structures would make the text more dynamic and easier to read.

5. The text could be more concise. Phrases like "in this paper, columnar anthracite is taken as the research object" could be simplified to "this study focuses on columnar anthracite.

Response: Thanks to the reviewer 's revision of the abstract part of the manuscript, I learned a novel way of writing. I think that modifying the abstract according to your requirements can make the writing focus of this article clearer and easier to understand, so that this article has been raised to a higher level. The abstracts modified according to the several amendments you proposed above are as follows :

The study of the adsorption characteristics of coal is of great significance to gas prevention and CO2 geological storage. To explore the adsorption mechanism of coal, this study focuses on columnar anthracite. Adsorption tests on coal rock under a range of physical field conditions were conducted using the volumetric method. The adsorption characteristics of anthracite for CO₂, CH₄, and N₂ gases under different conditions were investigated using Grand Canonical Monte Carlo (GCMC) and Molecular Dynamics (MD) methods,The results showed that the adsorption capacities of anthracite for these three gases are in the order of CO2 > CH4 > N2, and that the adsorption capacity increases with increasing gas injection pressure. The CO2/CH4/N2 gas molecule adsorption capacity of the anthracite macromolecular structure model decreases with increasing temperature. The increase in temperature has the greatest influence on the CO2 absorption capacity, followed by the CH4 and N2 adsorption capacities. The research offers a theoretical basis for the control of coal mine gas and the geological storage of CO2.

Introduction:

1. The introduction provides a thorough background on the significance of studying adsorption characteristics in coal, particularly anthracite. However, it could be condensed to improve readability and focus more directly on the study's objectives.

Response: Thank you for the guidance provided by the reviewers, and I agree with you very much. Therefore, I deleted two redundant references to make the background description of the adsorption characteristics of anthracite more concise.

2. While the introduction references prior studies, it could benefit from a more detailed comparison with previous research, especially in terms of how this study advances our understanding of the topic. Specifically, what gaps in the literature does this study aim to fill? To strengthen this section, I recommend citing recent studies that have explored similar topics in the context of carbon capture and storage (CCS) and methane recovery, such as those by https://doi.org/10.1016/j.envres.2023.116879 and https://doi.org/10.1016/j.jece.2023.110833. These articles provide relevant insights and comparisons that can help contextualize the novelty of your work, particularly in how your study contributes to the ongoing development of efficient adsorption technologies. Citing these sources will also demonstrate how your research builds upon and extends the current state of knowledge in this critical area.

Response: Thank the reviewers for helping me find two excellent references. I think these two articles have profound insights in the field of studying how to apply efficient technology to improve adsorption. After intensive reading of the two articles, I benefited a lot. Therefore, I added these two papers as references in my manuscript, as shown below :

[9] Ul W M ,Ul M S H ,Nasir S S , et al. Enhancing CO2 separation from N2 mixtures using hydrophobic porous supports immobilized with tributyl-tetradecyl-phosphonium chloride [P44414][Cl].[J].Environmental research,2023,237(P1):116879-116879. doi：10.1016/j.envres.2023.116879

[13] Tauqir A ,Saood M M ,Ullah S K , et al.Synthesis and Adsorptive Performance of a Novel Triazine Core-Containing Resin for the Ultrahigh Removal of Malachite Green from Water[J].Arabian Journal for Science and Engineering,2022,48(7):8571-8584. doi：10.1007/S13369-022-07015-W.

3. The introduction should highlight the novelty of the study more clearly. What distinguishes this study from previous work on similar topics? Is it the combination of experimental and simulation approaches, the specific conditions tested, or the focus on anthracite?

Response: As evidenced by the preceding analysis, research has been conducted on the adsorption characteristics of coal seams. Nevertheless, the majority of scholars have focused their research on pulverised coal, with relatively few studies examining columnar coal, which is in closer proximity to the c

---

## [Decision Letter · Decision Letter 2]

8 Sep 2024

The study on the adsorption characteristics of anthracite under different temperature and pressure conditions

PONE-D-24-04323R2

Dear Dr. Zhao,

We’re pleased to inform you that your manuscript has been judged scientifically suitable for publication and will be formally accepted for publication once it meets all outstanding technical requirements.

Kind regards,

Mashallah Rezakazemi

Academic Editor

PLOS ONE

Additional Editor Comments (optional):

Reviewers' comments:

Reviewer's Responses to Questions

**Comments to the Author**

1. If the authors have adequately addressed your comments raised in a previous round of review and you feel that this manuscript is now acceptable for publication, you may indicate that here to bypass the “Comments to the Author” section, enter your conflict of interest statement in the “Confidential to Editor” section, and submit your "Accept" recommendation.

Reviewer #3: All comments have been addressed

Reviewer #4: All comments have been addressed

Reviewer #5: All comments have been addressed

2. Is the manuscript technically sound, and do the data support the conclusions?

Reviewer #3: Yes

Reviewer #4: Yes

Reviewer #5: Yes

3. Has the statistical analysis been performed appropriately and rigorously? 

Reviewer #3: Yes

Reviewer #4: Yes

Reviewer #5: Yes

4. Have the authors made all data underlying the findings in their manuscript fully available?

Reviewer #3: Yes

Reviewer #4: Yes

Reviewer #5: Yes

5. Is the manuscript presented in an intelligible fashion and written in standard English?

Reviewer #3: Yes

Reviewer #4: Yes

Reviewer #5: Yes

6. Review Comments to the Author

Reviewer #3: The authors have made all changes recommended by the reviewer. I consider that the manuscript is ready for publication.

Reviewer #4: (No Response)

Reviewer #5: My decision is to accept this article after the authors have responded to the comments.Thank you for your submission .The comments and questions raised in the previous review are addressed thoroughly. Your revisions was directly respond to the feedback provided, enhancing the clarity, validity, and ethical soundness of the research.

7. PLOS authors have the option to publish the peer review history of their article (what does this mean? ). If published, this will include your full peer review and any attached files.

**Do you want your identity to be public for this peer review?** For information about this choice, including consent withdrawal, please see our Privacy Policy .

Reviewer #3: No

Reviewer #4: **Yes: ** Mohammad Hadi Nematollahi

Reviewer #5: **Yes: ** Dr MAHDI SHEIKH

---

## [Editor Report · Acceptance letter]

PONE-D-24-04323R2

PLOS ONE

Dear Dr. Zhao,

I'm pleased to inform you that your manuscript has been deemed suitable for publication in PLOS ONE. Congratulations! Your manuscript is now being handed over to our production team.

Kind regards,

on behalf of

Dr. Mashallah Rezakazemi

Academic Editor

PLOS ONE